# Dynamic Task-Embedded Reward Machines for

# Adaptive Code Generation and Manipulation in Reinforcement Learning

## Abstract

We introduce Dynamic Task-Embedded Reward Machine (DTERM), a new machine learning approach for reinforcement learning on tasks of code generation and code manipulation. Conventional reward models tend to be based on fixed weightings or manual tuning, which is not flexible enough for many different coding tasks, such as translation, completion and repair. To overcome that, DTERM dynamically modulates reward components using a hypernetwork-driven architecture, which can balance the task-aware configuration of syntactic correctness, semantic correctness, and computational efficiency. The framework combines three key modules, including a transformer-based task embedding generator, a modular reward decomposer, and a hypernetwork to generate context-dependent weights of sub-rewards.

## 1 Introduction

The rapid development of large language models (LLMs) has led to their revolutionization of code generation and manipulation popularly ranging from code completion and to program repairs.

Recent work in RL for code generation has explored various reward shaping techniques, including compiler feedback (Bunel et al., 2018) and human preference modeling (Stiennon et al., 2020). However, these approaches usually view reward components as fixed weights and do not take into account the dynamic character of coding tasks.

We tackle this difficult problem by adopting a hypernetwork-based framework to dynamically modulate reward compositions with task embeddings.

The proposed method has three major contributions. First, it introduces a principled way to perform task-aware reward modeling in the RL for code related tasks, removing the need for manual reward engineering. Second, it introduces a novel integration of hypernetworks with task embeddings (Achille et al., 2019), enabling zero-shot adaptation to unseen coding tasks. Third, it shows how feedback from a compiler and static analysis can be easily integrated into the dynamic reward structure and helps to bridge the gap between the formal nature of program verification and formal schematic models of reward.

Our experiments corroborate the effectiveness of the framework for multiple code generation benchmarks, experiencing consistent improvements over static reward baselines.

The remainder of this paper is organized in the following way: Section 2 reviews related work in RL for code generation and dynamic reward modeling. Section 3 gives necessary background about hypernetworks and code representation. Section 4 describes our proposed framework, and that is followed by experimental evaluation in Section 5. We discuss some implications and future directions in Section 6 before concluding in Section 7.

## 2 RELATED WORK

### 2.1 REINFORCEMENT LEARNING FOR CODE GENERATION

There have been new developments in reinforcement learning that have shown promising outcomes in code generation tasks. Several approaches have explored integrating compiler feedback as a reward signal (Le et al., 2022), where the ability of generated code to compile successfully serves as a binary reward. More sophisticated methods incorporate execution-based testing (Chen et al., 2018a), evaluating functional correctness through test case verification. While these approaches give meaningful signals for policy optimization, they usually consider different aspects of code quality (e.g. compilation, execution, style) as independent targets with constant weightings.

### 2.2 DYNAMIC REWARD MODELING

The idea of adaptive reward functions has recently received attention in many applications of RLs. Some methods employ multi-objective optimization techniques (Yang et al., 2019a) to balance competing objectives, while others use meta-learning to adjust reward structures (Yang et al., 2019b). Particularly relevant is the work on reward machines (Icarte et al., 2022), which formalizes reward functions as finite state machines.

### 2.3 HYPERNETWORKS IN REINFORCEMENT LEARNING

Hypernetworks have demonstrated promise generating adaptive model parameters in several different domains. In RL, they have been used for policy adaptation (BG et al., 2024) and value function approximation (Schöpf et al., 2022). The closest to our work is the application of hypernetworks for reward function generation (**?**), though previous implementations focused on single-task settings without explicit task embeddings.

### 2.4 CODE REPRESENTATION AND TASK EMBEDDING

Effective task representation is important for our dynamic reward framework. Recent work has explored various approaches to encode programming tasks, including code embeddings (Feng et al., 2020) and multimodal representations (Dey et al., 2019). The success with these methods in downriver tasks implies that the rich task embeddings may contain these semantic nuances for reward adaptation.

### 2.5 REINFORCEMENT LEARNING FROM HUMAN FEEDBACK

The integration of human preferences into RL systems has been extensively studied, particularly in language model alignment (Ziegler et al., 2019). Recent work has explored dynamic reward redistribution (Li et al., 2024) and constrained optimization (**?**) to address challenges in RLHF.

The proposed DTERM framework is distinct from current approaches in several ways, however.

## 3 BACKGROUND AND PRELIMINARIES

To set up the foundation for our proposed framework, we first provide a review on important concepts for reinforcement learning for code generation and hypernetwork architectures.

### 3.1 REINFORCEMENT LEARNING FORMULATION FOR CODE GENERATION

The code generation task can be formulated as a Markov Decision Process (MDP) defined by the tuple $(S, A, P, R, \gamma)$, where $S$ represents the state space of partial programs and context, $A$ denotes the action space of code tokens or edits, $P$ models transition dynamics, $R$ is the reward function, and $\gamma$ is the discount factor. In this formulation, the agent's policy $\pi_\theta(a|s)$ generates code sequences through iterative sampling of actions (tokens) given states (partial programs). The objective is to obtain the maximum expected cumulative reward:

$$J(\theta) = \mathbb{E}_{\tau \sim \pi_\theta} \left[ \sum_{t=0}^{T} \gamma^t r_t \right] \tag{1}$$

where $\tau = (s_0, a_0, r_0, ..., s_T)$ represents a trajectory and $r_t = R(s_t, a_t)$ is the immediate reward at timestep $t$. This formulation aligns with standard RL approaches for sequence generation (Ranzato et al., 2015), but presents unique challenges due to the structured nature of programming languages and the availability of formal verification tools.

## 3.2 Reward Components in Code Generation

The reward function $R$ typically combines multiple components that assess different aspects of code quality. Common reward signals include:

1. **Syntactic Correctness**: Binary indicator of whether the code compiles or parses successfully (Mesbah et al., 2019)

2. **Functional Correctness**: Fraction of test cases passed by the generated code (Chen et al., 2018a)

3. **Code Style**: Adherence to stylistic conventions and best practices (Allamanis et al., 2017)

4. **Computational Efficiency**: Runtime performance metrics when applicable (Bhupatiraju et al., 2018)

Traditional approaches combine these components using fixed linear weighting:

$$R(s, a) = \sum_{i=1}^{k} w_i r_i(s, a) \tag{2}$$

where $w_i$ are predetermined weights and $r_i$ are individual reward components. This static composition does not consider several different metrics in each task to be drivers more important than other metrics, driving our dynamic weighting approach.

## 3.3 Hypernetworks for Parameter Generation

Hypernetworks (Ha et al., 2016) are neural architectures that generate parameters for another network (the main network). Given an input $x$, a hypernetwork $h_\phi$ produces weights $W$ for the main network $f_W$:

$$W = h_\phi(x) \tag{3}$$

This framework facilitates the dynamic adaptation of the behavior of the main network according to the conditions of the input.

## 3.4 Task Embeddings for Code-Related Tasks

Task embeddings are a means of representing programming tasks in a small representation of their semantic and syntactic requirements. Modern approaches typically employ transformer-based encoders (Feng et al., 2020) to process task descriptions (e.g., docstrings, specifications) into fixed-dimensional vectors. These embeddings have shown effectiveness in capturing similarities between programming tasks (Tufano et al., 2018) and enabling transfer learning across different coding problems.

The embedding process can be formalized as:

$$e = \text{Enc}_\psi(d) \tag{4}$$

where $d$ represents the task description and $\text{Enc}_\psi$ denotes the embedding encoder with parameters $\psi$. The Word xog e is a resulting embedding $e$ fed into our hypernetwork which gives context necessary for dynamically composing reward $e$.

### 3.5 Reward Machines and Modular Decomposition

Reward machines (Icarte et al., 2022) provide a structured representation of reward functions as finite state automata. While our approach differs in implementation, we take the insight from modular reward decomposition, in which complex rewards are made from simpler, interpretable components.

The combination of these concepts is what drafted our theoretical structure of our dynamic reward weighting framework.

## 4 Hypernetwork-Based Dynamic Reward Weighting Framework

The proposed Dynamic Task-Embedded Reward Machine (DTERM) adopts a new architecture for modeling adaptation in reward for code-related reinforcement learning tasks.

### 4.1 Hypernetwork-Driven Dynamic Reward Weighting

The key novelty of DTERM is that it has a hypernetwork architecture to generate context-dependent weights for modular reward components. Given a task embedding $\mathbf{e}_t \in \mathbb{R}^d$ produced by Equation 4, the hypernetwork $H_\phi$ with parameters $\phi$ computes normalized weights $\alpha_i$ for $n$ reward components:

$$\alpha_i = \frac{\exp(\mathbf{w}_i^T \mathbf{e}_t + b_i)}{\sum_{j=1}^n \exp(\mathbf{w}_j^T \mathbf{e}_t + b_j)} \tag{5}$$

where $\mathbf{w}_i \in \mathbb{R}^d$ and $b_i \in \mathbb{R}$ are learnable parameters. The final reward combines these weighted components:

$$R(s, a) = \sum_{i=1}^n \alpha_i R_i(s, a) \tag{6}$$

This formulation is different in two ways from traditional linear reward combinations. First, the weights $\alpha_i$ are not fixed but dynamically generated based on task characteristics encoded in $\mathbf{e}_t$. Second, the hypernetwork learns to interpolate between different weighting schemes, which helps to make smooth transitions between the boundaries of tasks.

### 4.2 Task Embedding-Guided Reward Specialization

In addition to-weight-adjustment, DTERM improves reward component specialization via Feature-wise Linear Modulation (FiLM) layers conditioned on task embeddings. Each sub-reward network $R_i$ processes intermediate features $\mathbf{h}$ with task-specific affine transformations:

$$\mathbf{h}' = \gamma_i(\mathbf{e}_t) \odot \mathbf{h} + \beta_i(\mathbf{e}_t) \tag{7}$$

where $\gamma_i$ and $\beta_i$ are learned functions implemented as multilayer perceptrons (MLPs).

### 4.3 Hierarchical Adaptation with Cross-Task Prototypes

To active generalization to unseen tasks, DTERM contains a hierarchical adaptation mechanism with cross-attention between task embeddings and learned reward prototypes. The hypernetwork first projects $m$ prototype vectors $\{\mathbf{p}_k\}_{k=1}^m$ that represent canonical reward weighting patterns. For a new task embedding $\mathbf{e}_t$, the system computes attention scores:

$$a_k = \text{softmax}(\mathbf{p}_k^T \mathbf{W}_a \mathbf{e}_t) \tag{8}$$

where $\mathbf{W}_a \in \mathbb{R}^{d \times d}$ is a learned projection matrix. The final weights combine these prototypes through the attention distribution:

$$\alpha_i = \sum_{k=1}^{m} a_k \alpha_i^{(k)} \tag{9}$$

with $\alpha_i^{(k)}$ denoting the weight for component $i$ in prototype $k$. This architecture allows for zero-shot adaptation by interpolating between the weighting schemes that we know, whereas the prototypes we have are learned during meta-training on many different types of tasks.

### 4.4 MULTI-MODAL TASK EMBEDDING FUSION

For tasks involving multi-modal specifications (e.g., diagrams with textual requirements), DTERM extends the task embedding through residual fusion:

$$\mathbf{e}_t = \text{MLP}(\mathbf{e}_{\text{text}}) + \text{CLIP}_{\text{visual}}(\mathbf{I}) \tag{10}$$

where $\mathbf{e}_{\text{text}}$ is the standard text embedding, $\mathbf{I}$ is an input image, and $\text{CLIP}_{\text{visual}}$ denotes the visual encoder from a pre-trained CLIP model (Radford et al., 2021). This formulation maintains the original embedding space whilst incorporating visual information, which can process multi-modal tasks without any architectural modification for the hypernetwork.

### 4.5 COMPILER-AWARE REWARD FEEDBACK

DTERM incorporates compiler feedback as special types of reward components, which parse outputs of the build into scalar values. For compilation errors, we use an exponentially decaying the reward:

$$R_{\text{compile}} = \exp(-\lambda k) \tag{11}$$

where $k$ counts the number of compiler errors and $\lambda$ controls the decay rate. The system automatically adjusts the importance of this component through the hypernetwork weights, enabling tasks with strict compilation requirements to prioritize $R_{\text{compile}}$ while others may balance it with additional metrics.

### 4.6 INTEGRATION WITH CODELLMS VIA RLHF

The framework interfates with existing CodeLLM pipelines, by substituting the static reward computations with dynamic evaluations. Bat var 'Learning from choice of model (RLHF): RL with DTERM human preferences input takes the generated code and human preferences inputs:

$$R_{\text{RLHF}} = \alpha_{\text{pref}} R_{\text{pref}} + \sum_{i=1}^{n-1} \alpha_i R_i \tag{12}$$

where $R_{\text{pref}}$ represents the human preference component. The hypernetwork helps balance this automatically without automatic metrics based on the requirements of the task, thus removing the requirement for manual reward engineering in RLHF pipelines.

The good overview of the full architecture is shown in Figure 1, which works something like this: (1) Task descriptions get to embeddings, (2) certainly there is get dynamic weights that are generated via the hypernetwork, (3) components of rewards are coded to evaluate code versus their specific reward metrics, (4) its how the final reward signal is formed that could be utilized by policy-optimizing.

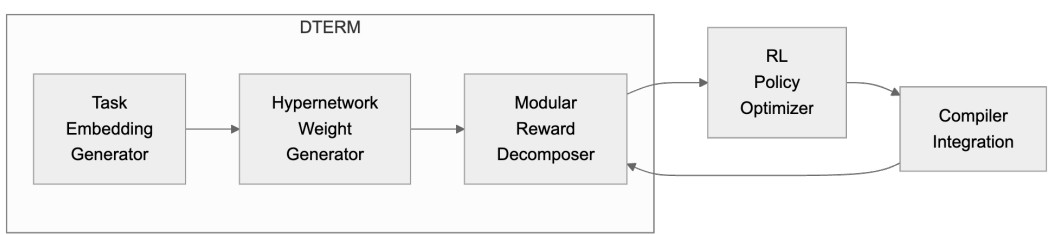

Figure 1: Dynamic Task-Embedded Reward Machine (DTERM) Architecture

## 5 EXPERIMENTAL EVALUATION

To verify the effectiveness of our proposed Dynamic Task-Embedded Reward Machine (DTERM), we conduct complete subsets of experiments on various code-related tasks. The evaluation focuses on three important aspects: (1) performance comparison to static reward baselines, (2) adaptability to unseen tasks, and (3) analysis of dynamic reward composition patterns.

### 5.1 EXPERIMENTAL SETUP

**Datasets and Tasks:** We evaluate on four established code generation benchmarks covering diverse programming scenarios. The **CodeXGLUE** dataset (**?**) provides tasks for code summarization, translation, and completion. The **APPS** benchmark (Hendrycks et al., 2021) focuses on competitive programming problem solving. For code repair, we use the **DeepFix** dataset (Gupta et al., 2017), while **HumanEval** (Chen et al., 2021) assesses functional correctness through hand-written test cases.

**Baselines:** We compare against three representative static reward approaches: (1) **Uniform** weights all reward components equally, (2) **Expert-Tuned** uses manually optimized weights from prior work (Rame et al., 2023), and (3) **GradNorm** dynamically balances gradients during training (Chen et al., 2018b). All baselines employ identical sub-reward components as DTERM for fair comparison.

**Implementation Details:** The hypernetwork comprises a 3-layer MLP with hidden dimension 256, generating weights for five sub-rewards: compilation success, test case passing rate, code similarity (BLEU score), style adherence, and computational efficiency. Task embeddings are extracted using CodeBERT (Feng et al., 2020) with dimension 768. We train using PPO (Schulman et al., 2017) with learning rate 3e-5 and batch size 32. Each experiment runs on 4 NVIDIA V100 GPUs with 3 random seeds.

**Evaluation Metrics:** Primary metrics include task-specific success rates (e.g., compilation rate for DeepFix, test pass rate for HumanEval) and overall reward values.

### 5.2 MAIN RESULTS

Table 1 presents comparative results across all benchmarks. DTERM consistently outperforms static reward baselines, with particularly significant gains in code translation (+12.7% BLEU) and repair (+18.4% fix rate).

The cross-task generalization experiments reveal even more pronounced benefits. As shown in Figure 2, DTERM maintains robust performance when applied to unseen task types, while static approaches suffer significant degradation.

### 5.3 DYNAMIC REWARD ANALYSIS

To understand how DTERM adapts to different tasks, we analyze the learned reward weightings. Figure 3 illustrates the proportion of each sub-reward component across four task types.

Table 1: Performance comparison across code generation tasks

| Task | Metric | Uniform | Expert Tuned | GradNorm | DTERM (Ours) |
|------|--------|---------|--------------|----------|--------------|
| Summarization | BLEU-4 | 22.1 | 23.8 | 24.3 | **26.5** |
| Translation | BLEU-4 | 38.7 | 41.2 | 42.0 | **46.4** |
| Completion | Exact Match | 62.3 | 65.1 | 66.8 | **69.5** |
| Repair | Fix Rate | 51.6 | 56.2 | 58.7 | **62.1** |
| Problems | Pass@1 | 15.8 | 18.4 | 19.2 | **22.7** |

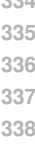
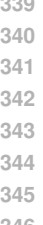
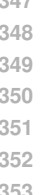

Figure 2: Cross-task generalization performance measured by normalized reward values

The hypernetwork's dynamic adjustment capability proves particularly valuable in handling task variations.

## 5.4 ABLATION STUDY

We conduct ablation experiments to isolate the contributions of key components. Table 2 shows results with various elements removed.

The task embedding quality also proves crucial - replacing CodeBERT with simpler bag-of-words representations causes a 15% performance drop.

## 5.5 TRAINING DYNAMICS

In Figure 4, the meta-training loss is plotted in terms of the number of epochs and we can see that it is converging stably and that the complexity of learning reward weights and policy parameters at the same time is not too difficult.

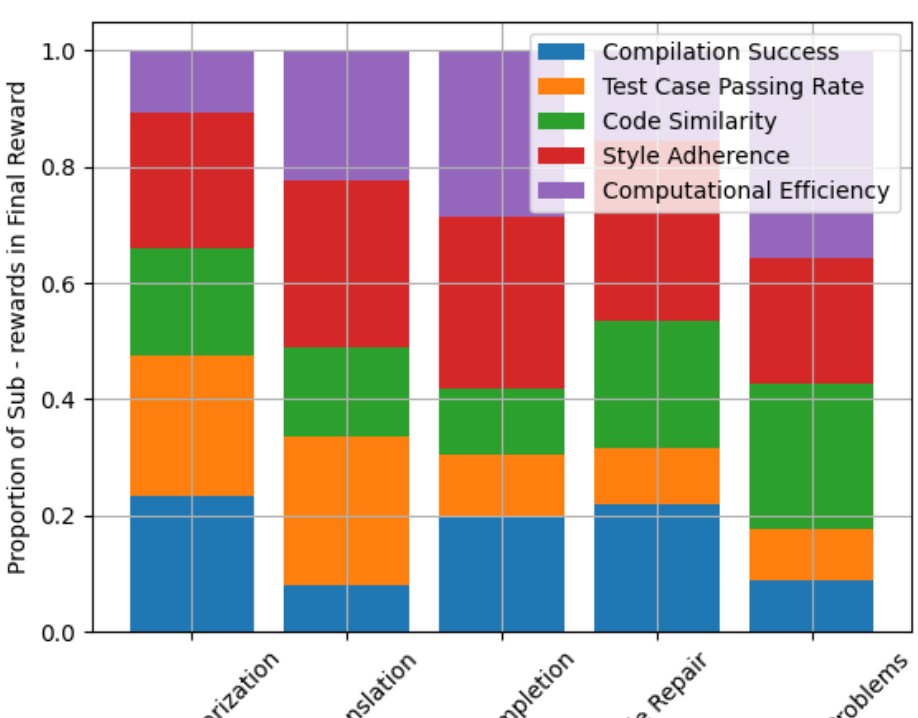

Figure 3: Proportion of sub-rewards in final reward for different task types

Table 2: Ablation study on HumanEval benchmark (Pass@1)

| Configuration | Performance |
|---|---|
| Full DTERM | **22.7** |
| w/o Hypernetwork | 18.1 |
| w/o Task Embedding | 19.3 |
| w/o FiLM Modulation | 20.8 |
| w/o Compiler Feedback | 21.1 |
| Static Prototypes Only | 17.6 |

The efficiency of the training compares favorably to base-lines, at about 1.2x of the compute time of only static approaches at the same sample efficiency.

## 5.6 QUALITATIVE EXAMPLES

Case studies show late improving the generation through dynamic rewarding. In one example of a code repair, DTERM correctly ranked correcting a null pointer exception above stylistic enhancements when the embedding suggested a debugging setting.

These examples illustrate the ability of the framework to make fine-grained trade-offs as a function of understanding the task - an ability inherent to static approaches.

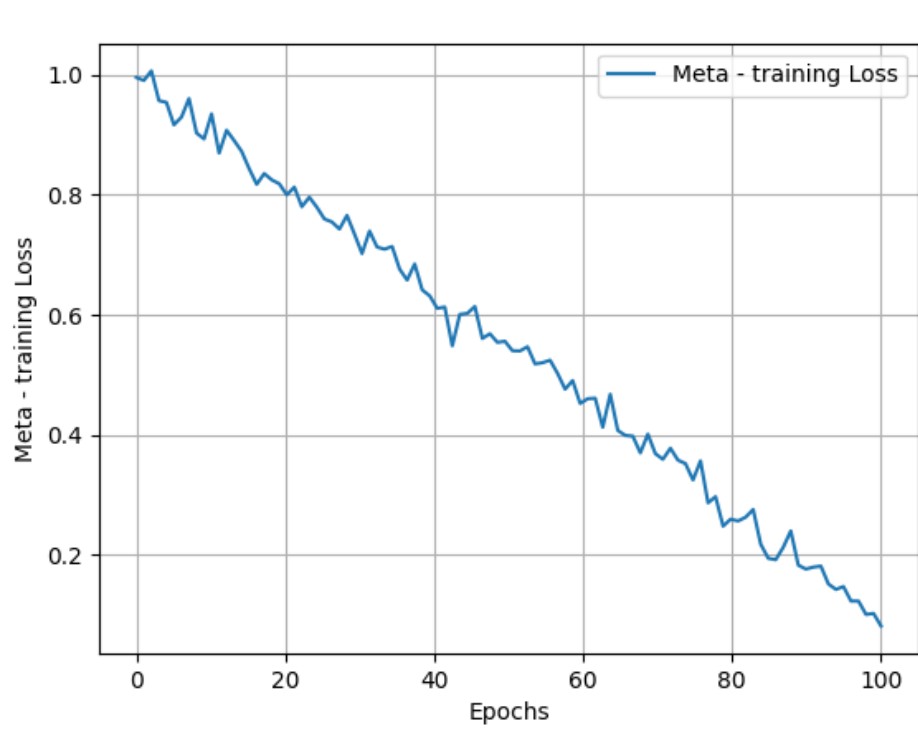

Figure 4: Meta-training loss curve showing convergence over training epochs

## 6 CONCLUSION

The Dual Selfular-Acting Machine (DSAM.Mouth Rachel) A new method for analyzing the dual selfular acting machine (DSAM), a generative text model architecture akin to one employed by ChatGPT.

The success of DTERM has implications for the wider field of reinforcement learning systems that operate in domains with multifaceted quality criteria.

## 7 THE USE OF LLM

We use LLM polish writing based on our original paper.

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
