# OpenReview forum: "Dynamic Task-Embedded Reward Machines for \\ Adaptive Code Generation and Manipulation \\ in Reinforcement Learning"
_ICLR.cc/2026/Conference — Submitted to ICLR 2026_

### Official Review · Reviewer_oG3A · 2025-10-30

**Soundness:** 1
**Presentation:** 1
**Contribution:** 1
**Rating:** 0
**Confidence:** 4

**Summary:**

The manuscript strongly resembles AI-generated content and may have been produced as an internal test for prospective AI researchers. If so, it suggests that the current state of such roles remains immature and requires further development.

**Strengths:**

I believe this paper was generated by AI. If not, please let me know.

**Weaknesses:**

I believe this paper was generated by AI. If not, please let me know.

**Questions:**

See Weaknesses.

---

### Official Review · Reviewer_NmKe · 2025-10-31

**Soundness:** 1
**Presentation:** 1
**Contribution:** 1
**Rating:** 2
**Confidence:** 1

**Summary:**

This paper introduces Dynamic Task-Embedded Reward Machines (DTERM), a novel framework for reinforcement learning (RL) in code generation and manipulation tasks. Unlike traditional reward models that rely on fixed or manually tuned weights, DTERM employs a hypernetwork-driven architecture to dynamically adjust the contributions of various reward components - such as syntactic correctness, semantic correctness, and computational efficiency - based on task embeddings. The framework integrates a transformer-based task embedding generator, a modular reward decomposer, and a hypernetwork to produce context-aware reward weightings. Experiments across multiple benchmarks (e.g., CodeXGLUE, APPS, DeepFix, HumanEval) demonstrate consistent improvements over static reward baselines and strong generalization to unseen tasks.

**Strengths:**

The one potential strength of the work is the identification of a valid problem: static reward functions are indeed a limitation in RL for code generation. The idea of making them adaptive is a worthwhile direction to explore.

**Weaknesses:**

The weaknesses are severe and fundamental.

Fatally Compromised Presentation: The numerous grammatical errors, incoherent sentences, and missing content (like Eq. 4 and Fig. 1) make the paper unreadable and un-reviewable in its current state. This alone warrants rejection.

Unsupported Claims: All major claims regarding performance and generalization are made without the necessary statistical evidence or rigorous experimental design to support them.

Technical Debt: The model is complex (hypernetwork, task encoder, multiple reward modules, FiLM layers, prototype attention), yet the paper provides no analysis of computational cost, training stability, or sensitivity to hyperparameters.

**Questions:**

The paper is riddled with grammatical errors and incomplete sentences (e.g., Sec 4.6, Sec 6). Can the authors provide a coherent, fully proofread version that accurately represents their work?

Where is Equation 4 and the detailed architecture diagram (Figure 1)? The current manuscript is incomplete without them.

Where are the results of statistical significance tests (e.g., p-values) for the performance improvements reported in Table 1? Can the authors prove their method's advantage is not due to variance?

The "unseen tasks" in the generalization experiment are not defined. What are these tasks, and how do they semantically differ from the training tasks? Please provide concrete examples and task-level success metrics, not just normalized reward.

The conclusion (Section 6) describes a completely different model ("Dual Selfular-Acting Machine"). Has the manuscript been compromised during submission? Please clarify this critical discrepancy.

---

### Official Review · Reviewer_p2jD · 2025-10-31

**Soundness:** 1
**Presentation:** 1
**Contribution:** 1
**Rating:** 0
**Confidence:** 4

**Summary:**

The paper presents DTERM, a framework for RL in code generation and manipulation tasks. DTERM combines transformer-based task embeddings, modular decomposition of reward components, and a hypernetwork that produces context-dependent weights over these components. Experiments across four prominent code-generation benchmarks show that DTERM outperforms static and manually tuned reward baselines, particularly in cross-task generalization and adaptability.

**Strengths:**

The paper addresses a relevant problem in RL for code generation, proposing a principled yet straightforward framework for dynamic reward weighting.

**Weaknesses:**

The paper lacks sufficient comparison with prior adaptive reward modeling work, making the novelty claims less convincing. The approach is heavily dependent on CodeBERT, with minimal analysis of robustness or generality. Moreover, there are noticeable writing and editing issues.

**Questions:**

No questions.

---

### Official Review · Reviewer_YZfp · 2025-11-12

**Soundness:** 1
**Presentation:** 1
**Contribution:** 1
**Rating:** 2
**Confidence:** 4

**Summary:**

This paper proposes Dynamic Task-Embedded Reward Machines (DTERM), a hypernetwork-based framework for dynamically weighting reward components in reinforcement learning for code generation and manipulation tasks. Instead of using static weights for sub-rewards (e.g., syntax, functionality, style), DTERM employs task embeddings derived from transformer encoders (e.g., CodeBERT) to generate adaptive weighting through a hypernetwork. The method is tested on several code generation benchmarks such as CodeXGLUE, HumanEval, APPS, and DeepFix, reporting modest improvements over static baselines.

**Strengths:**

While the motivation of dynamic reward composition is reasonable, the technical soundness is weak. The method lacks clear theoretical grounding, ablation rigor, and reproducibility. Several equations are poorly defined, notation inconsistent, and experimental details insufficient to support the claims. The “cross-task prototype” and “FiLM modulation” mechanisms are mentioned but not rigorously formulated or justified. The last section (“Dual Selfular-Acting Machine”) seems unrelated and possibly mistakenly copied.

**Weaknesses:**

The idea of task-conditioned dynamic reward weighting via hypernetworks is moderately interesting, but not novel. Prior work on meta-RL, reward machines, and multi-objective RL (e.g., Icarte et al., 2022; Yang et al., 2019a,b) already explore similar concepts with stronger theoretical and experimental grounding. The paper lacks a substantial new insight or methodological advance. The improvements in Table 1 (~2–4%) are minor and may be within variance.

**Questions:**

What exactly is the learning objective of the hypernetwork? Is it trained jointly with the policy or separately?

How are task embeddings obtained—are they frozen or fine-tuned during RL training?

Why does the “Dual Selfular-Acting Machine” appear in the conclusion—was this an editing error?

How does DTERM differ in practice from meta-learning reward-weight modulation (e.g., GradNorm or MAML-style adaptation)?

Were any of the experiments verified for statistical significance or reproducibility?

---

### Meta-Review · Area_Chair_L9vS · 2026-01-11

**Summary:**

The reviewers unanimously rated this submission poorly across all dimensions (Soundness, Presentation, and Contribution all received 1/poor from every reviewer). The concerns fall into several categories of escalating severity.

Manuscript Integrity and Completeness Issues. Multiple reviewers identified that the conclusion section describes an entirely different system called "Dual Selfular-Acting Machine," which appears unrelated to the DTERM framework presented in the paper. Reviewer NmKe explicitly asks whether the manuscript was "compromised during submission." Additionally, Equation 4 and Figure 1 are reportedly missing from the manuscript. Reviewer oG3A directly states a belief that the paper was AI-generated and requests clarification if this assessment is incorrect.

Presentation Quality. All reviewers cite severe presentation problems including grammatical errors, incoherent sentences, poorly defined equations, and inconsistent notation throughout the manuscript. Reviewer NmKe characterizes these issues as "fatally compromised presentation" that makes the paper "unreadable and un-reviewable."

Technical and Experimental Weaknesses. Reviewers note the absence of statistical significance testing for the reported improvements (2-4%), which may fall within experimental variance. The paper lacks rigorous ablation studies, computational cost analysis, and hyperparameter sensitivity analysis. Key mechanisms such as "cross-task prototype" and "FiLM modulation" are mentioned but not rigorously formulated. There is insufficient comparison with prior work on meta-RL, reward machines, and multi-objective RL that addresses similar problems.

Limited Novelty. Reviewers YZfp and p2jD both indicate that task-conditioned dynamic reward weighting via hypernetworks, while reasonable in motivation, is not sufficiently novel given existing literature in the area.

**Reviewer Concerns:**

Seems that no rebuttal was provided.

**Reviewer Scores:**

Seems that no rebuttal was provided.

---

### Decision · Program_Chairs · 2026-01-26

Reject